# Effective Orifice Area of Balloon-Expandable and Self-Expandable Transcatheter Aortic Valve Prostheses: An Echo Doppler Comparative Study

**DOI:** 10.3390/jcm10020186

**Published:** 2021-01-07

**Authors:** Mohamad Kanso, Marion Kibler, Sebastien Hess, Jérome Rischner, Philoktimon Plastaras, Michel Kindo, Minh Hoang, Fabien De Poli, Pierre Leddet, Hélène Petit, Floriane Zeyons, Annie Trinh, Kensuke Matsushita, Olivier Morel, Patrick Ohlmann

**Affiliations:** 1Pôle d’Activité Médico-Chirurgicale Cardiovasculaire, Hôpitaux Universitaires de Strasbourg, 67000 Strasbourg, France; mohamad.kanso@chru-strasbourg.fr (M.K.); marion.kibler@chru-strasbourg.fr (M.K.); sebastien.hess@chru-strasbourg.fr (S.H.); michel.kindo@chru-strasbourg.fr (M.K.); tam.hoangminh@chru-strasbourg.fr (M.H.); Helene.PETIT-EISENMANN@chru-strasbourg.fr (H.P.); floriane.zeyons@chru-strasbourg.fr (F.Z.); annie.trinh@chru-strasbourg.fr (A.T.); olivier.morel@chru-strasbourg.fr (O.M.); 2Hôpital Albert Schweitzer, 201 Avenue d’Alsace, 68003 Colmar, France; jerome.rischner@diaconat-mulhouse.fr (J.R.); philoktimon.plastaras@diaconat-mulhouse.fr (P.P.); 3Centre Hospitalier d’Haguenau, 64 Avenue Du Professeur Leriche, 67504 Haguenau, France; fabien.depoli@ch-haguenau.fr (F.D.P.); pierre.leddet@ch-haguenau.fr (P.L.)

**Keywords:** TAVI, Edwards Sapien 3, CoreValve, effective orifice area, transvalvular gradient, Doppler echocardiography

## Abstract

Published data on the size-specific effective orifice area (EOA) of transcatheter heart valves (THVs) remain scarce. Here, we sought to investigate the intra-individual changes in EOA and mean transvalvular aortic gradient (MG) of the Sapien 3 (S3), CoreValve (CV), and Evolut R (EVR) prostheses both at short-term and at 1-year follow-up. The study sample consisted of 260 consecutive patients with severe aortic stenosis who underwent transcatheter aortic valve implantation (TAVI). EOAs and MGs were measured with Doppler echocardiography for the following prostheses: S3 23 mm (*n* = 74; 28.5%), S3 26 mm (*n* = 67; 25.8%), S3 29 mm (*n* = 20; 7.7%), CV 23 mm (*n* = 2; 0.8%), CV 26 mm (*n* = 15; 5.8%), CV 29 mm (*n* = 24; 9.2%), CV 31 mm (*n* = 9; 3.5%), EVR 26 mm (*n* = 22; 8.5%), and EVR 29 mm (*n* = 27; 10.4%). Values were obtained at discharge, 1 month, 6 months, and 1 year from implantation. At discharge, EOAs were larger and MGs lower for larger-size prostheses, regardless of being balloon-expandable or self-expandable. In patients with small aortic annulus size, the hemodynamic performances of CV and EVR prostheses were superior to those of S3. However, we did not observe significant differences in terms of all-cause mortality according to THV type or size. Both balloon-expandable and self-expandable new-generation THVs show excellent hemodynamic performances without evidence of very early valve degeneration.

## 1. Introduction

Aortic stenosis (AS)—the most prevalent form of single native valve disease in European patients referred for the management of valvular heart disease [1]—poses a significant public health burden. Surgical aortic valve replacement remains the definitive treatment for severe symptomatic AS. However, transcatheter aortic valve implantation (TAVI) has emerged as the procedure of choice for patients who face a high risk of complications after surgery (e.g., in presence of advanced age or multiple comorbidities) [2].

Compared to surgical bioprostheses, first-generation percutaneous prostheses are capable of preventing patient–prosthesis mismatch (PPM) and are characterized by superior hemodynamic performances in terms of transprosthetic gradient—albeit at the expense of an increased aortic regurgitation [3].

The effective orifice area (EOA) measured by Doppler echocardiography is generally dependent on the prosthesis size, regardless of its positioning with either surgery [4] or TAVI [5].

However, there are few published data about reference hemodynamic parameters and their intra-individual changes that may occur over time in patients treated with new-generation valves (i.e., the self-expandable CoreValve^®^ (Medtronic, Minneapolis, MN, USA), the CoreValve Evolut^®^ R system (Medtronic), and the balloon-expandable Edwards Sapien 3^®^ system (Edwards Lifesciences Inc., Irvine, CA, USA). We therefore designed the current study to investigate this issue by focusing on the EOA and mean transvalvular aortic gradient (MG) in relation to valve size. We also examined the evolution of these parameters at 30-day and 1-year follow-up.

## 2. Materials and Methods

### 2.1. Study Sample

Patients with severe symptomatic AS who were treated with TAVI at our institution between December 2014 and September 2016 were deemed eligible for inclusion. All patients were considered to have a moderate-to-high surgical risk according to common risk scores and other clinical parameters (e.g., frailty, presence of porcelain aorta, and sequelae of chest irradiation). Patients were excluded if they were undergoing TAVI because of a degenerated aortic bioprosthesis (valve-in-valve procedure) or an extra-aortic valve treatment (i.e., mitral or tricuspid valve). Patients who died during the periprocedural period were not included. Three different types of prosthesis were investigated in the current study: (1) Edwards Sapien 3^®^ (S3), (2) Medtronic Corevalve^®^ (CV), and (3) Medtronic CoreValve Evolut^®^ R (EVR). The optimal prosthesis size was selected based on anatomical data obtained from computed tomography (CT), which was performed according to the current recommendations for CT imaging before TAVI [6]. All of the study patients were included in the French TAVI registry and provided their written informed consent.

### 2.2. Study Design

This was a single-center retrospective cohort study. The primary objective was to compare the intra-individual changes in EOA and MG of the S3, CV, and EVR prostheses at different time points (discharge, 1 month, 6 months, and 1 year from implantation). All analyses were conducted in relation to valve size.

### 2.3. Echocardiography and Doppler Measurements

Standard transthoracic echocardiography was performed at discharge and at different follow-up points according to the American Society of Echocardiography guidelines. All measurements were performed by a single observer who had extensive experience in echocardiography. Left ventricle ejection fraction (LVEF) was estimated using the biplane method of disks (modified Simpson’s rule). EOA and Doppler velocity index (DVI) were calculated using the continuity equation, whereas the left ventricular outflow tract (LVOT) was conventionally measured immediately below the annular plane. Because this approach is the most accurate for calculation of EOA in balloon-expandable valves [7], the same preimplantation LVOT diameter was used for subsequent examinations in most cases. When the measured LVOT during follow-up was different from the pre-TAVI LVOT value, EOA was recalculated using the preimplantation diameter in an effort to reduce the interobserver variability and unreliable estimations [8]. Because the flow acceleration phenomenon in a prosthetic valve can lead to overestimation of both EOA and the Doppler velocity index (DVI), the Doppler sample volume was placed immediately before the prosthesis to obtain a correct LVOT velocity time integral (VTI) spectrum [9]. The transaortic MG was estimated with the Bernoulli modified equation [10]. In order to detect PPM, EOA was indexed to the body surface area (BSA). The stroke volume index (SVi) was obtained using the VTI over both LVOT and BSA.

### 2.4. Selection of Prosthesis Size and Preinterventional CT

The prosthesis size was selected based on (1) the recommendation charts provided by the manufacturers and (2) the following preinterventional CT measurements: aortic annular area (CT area) for S3 valves and annular perimeter (P) for both CV and EVR prostheses. CT measurements were obtained using the osiriX software (Pixmeo SARL, Geneva, Switzerland) and/or the 3mensio Valves software (version 5.1.sp1; 3mensio Pie Medical Imaging, Maastricht, The Netherlands).

### 2.5. Endpoints

The primary endpoint was the EOA following TAVI (i.e., during the in-hospital phase). Secondary endpoints included the post-implantation transaortic MG, as well as the EOA and transaortic MG at 1 month, 6 months, and 1 year.

### 2.6. Statistical Analysis

Continuous variables are expressed as means ± standard deviations and compared with analysis of variance (ANOVA) followed by Bonferroni’s post hoc tests. Categorical variables are given as frequencies and percentages and analyzed with the chi-squared test or the Fisher’s exact test (as appropriate). All-cause mortality curves were plotted with the Kaplan–Meier method and compared with the log-rank test. All calculations were performed with the SPSS statistical package, version 11.5 (SPSS Inc., Chicago, IL, USA). Two-tailed *p* values < 0.05 were considered statistically significant.

## 3. Results

Between December 2014 and May 2016, a total of 249 consecutive patients underwent TAVI in our center. After the exclusion of early post-procedural deaths (*n* = 9), aortic valve-in-valve procedures (*n* = 8), and clinical indications other than severe AS (severe aortic regurgitation, *n* = 5; severe tricuspid regurgitation, *n* = 1), 226 patients were deemed eligible. In order to increase the sample size for certain specific prostheses (CV 31 mm, EVR 26 mm, and EVR 29 mm), we screened for eligibility another group of 86 consecutive patients who were treated with TAVI between May 2016 and September 2016. Of them, 37 patients underwent TAVI using one of the three prostheses of interest. After the exclusion of two patients who died early after the procedure and one case of severe aortic regurgitation, an additional 34 cases were included in the study—resulting in a final study sample of 260 patients.

The distribution of implanted prostheses was as follows: S3 23 mm, 74 (28.5%) patients; S3 26 mm, 67 (25.8%) patients; S3 29 mm, 20 (7.7%) patients; CV 23 mm, 2 (0.8%) patients; CV 26 mm, 15 (5.8%) patients; CV 29 mm, 24 (9.2%) patients; CV 31 mm, 9 (3.5%) patients; EVR 26 mm, 22 (8.5%) patients; and EVR 29 mm, 27 (10.4%) patients.

The baseline characteristics of the study patients are summarized in Table 1 and Table 2. Smaller TAVI devices (S3 23 mm, CV 26 mm, and EVR 26 mm) were more frequently implanted in women than in men and in patients with smaller body surface areas. There were no major intergroup differences in terms of clinical variables. Notably, LVEF was significantly lower in patients who received an S3 26 mm device compared with an EVR 26 mm prosthesis (*p* = 0.037). However, the indexed stroke volume did not differ significantly in the two groups. Patients with peripheral artery disease (PAD) were over-represented among those who were implanted with CV prosthesis.

### 3.1. EOA, Transaortic MG, and LVEF

The mean EOA at discharge was 2.0 ± 0.6 cm^2^ and increased in parallel with the prosthesis size (Table 3 and Figure 1) as follows: 1.6 ± 0.3 cm^2^ for S3 23 mm (*n* = 74); 1.9 ± 0.4 cm^2^ for S3 26 mm (*n* = 67); 2.4 ± 0.7 cm^2^ for S3 29 mm (*n* = 20); 1.5 ± 0.3 cm^2^ for CV 23 mm (*n* = 2); 2.3 ± 0.7 cm^2^ for CV 26 mm (*n* = 15); 2.7 ± 0.8 cm^2^ for CV 29 mm (*n* = 24); 1.9 ± 0.5 cm^2^ for CV 31 mm (*n* = 9); 2.2 ± 0.3 cm^2^ for EVR 26 mm (*n* = 22); and 2.4 ± 0.4 cm^2^ for EVR 29 mm (*n* = 27). EOA values within each subgroup remained stable at follow-up, the only exception being the CV 23 mm and CV 31 mm subgroups—for which the small sample sizes did not allow reliable assessments.

Abbreviations: LVEF, left ventricular ejection fraction; MG, transvalvular aortic mean gradient; SVi, stroke volume indexed to the body surface area; AVA, aortic valve area; AR, aortic regurgitation; MR, mitral regurgitation; LVOT, left ventricular outflow tract; Area-derived diam, aortic annulus area-derived diameter; Ann area, annulus area; Ann perim, annulus perimeter; AU, Agatston units

At discharge, the transaortic MG was found to differ significantly among different prostheses as follows: 11.74 ± 3.45 mmHg for S3 23 mm (*n* = 74); 9.59 ± 3.63 mmHg for S3 26 mm (*n* = 67); 8.95 ± 2.94 mmHg for S3 29 mm (*n* = 20); 14.5 ± 6.36 mmHg for CV 23 mm (*n* = 2); 6.70 ± 3.97 mmHg for CV 26 mm (*n* = 15); 6.40 ± 3.22 mmHg for CV 29 mm (*n* = 24); 8.57 ± 4.77 mmHg for CV 31 mm (*n* = 9); 8.77 ± 3.22 mmHg for EVR 26 mm (*n* = 22); and 7.48 ± 3.57 mmHg for EVR 29 mm (*n* = 27). Despite differences in EOAs, MG at follow-up remained stable across different sizes of the same valve, the only exception being S3 23 mm—whose MG was higher than that of S3 26 mm (*p* = 0.013) and also slightly increased between the first and the sixth month after implantation (*p* = 0.022).

### 3.2. Patient–Prosthesis Mismatch

Severe aortic valve mismatch (defined as an AVAi < 0.65 cm^2^/m^2^) was observed in six (2.4%) patients. Specifically, four cases occurred with S3 23 mm, one with S3 26 mm, and one with CV 31 mm. Moderate aortic valve mismatch (defined as an AVAi < 0.85 cm^2^/m^2^) occurred in 39 (16%) patients. Most of these cases (*n* = 24; 33.6%) were observed with S3 23 mm (*p* < 0.001), followed by CV 26 mm (*n* = 1; 6.7%), and CV 31 mm (*n* = 3; 33.3%).

### 3.3. Impact of Prosthesis Size on the Hemodynamic Characteristics

EOAs in patients with the same aortic annulus range were compared. Patients were categorized in three groups according to the size of their annulus on CT images and the corresponding Sapien S3 charts as follows: small (314–430 mm^2^), medium (430−540 mm^2^), and large (546−700 mm^2^) annulus. These categories were selected in an effort to make them fit to the sizing table of the Sapien S3 valve (Figure 2). We observed significant larger EOAs in patients implanted with the CoreValve and Evolut R bioprostheses compared to the Sapien S3 valve. This was especially evident for the small annulus group and, to a lesser extent, for the medium annulus group. No significant difference was observed in the large annulus group.

### 3.4. Paravalvular Aortic Regurgitation

With the exception of one patient with moderate-to-severe paravalvular aortic regurgitation, post-procedural regurgitation at follow-up was either absent or mild to moderate. Specifically, moderate paravalvular aortic regurgitation at discharge was observed in 11.9% of the study patients. This rate decreased to 5% at the 6-month and 1-year assessments, without significant intergroup differences (*p* = 0.772 at 1-year follow-up).

### 3.5. Survival Analysis

A total of 271 patients were included in an intention-to-treat survival analysis (Figure 3). At 1-year follow-up, there were 27 deaths (10.2%). Five patients were lost to follow-up. Among the 27 patients who died, the distribution of the implanted prostheses was as follows: CV, *n* = 9 (17%); EVR, *n* = 2 (4.9%); and S3, *n* = 16 (9.2%). There were no significant differences in terms of all-cause mortality between different prostheses (log-rank test; *p* = 0.101).

## 4. Discussion

TAVI prostheses are hemodynamically superior to surgical bioprostheses. Compared with both stented and stentless surgical valves, lower MGs, larger EOAs, and lower PPM rates have been reported for first-generation Edwards prostheses, albeit at the expense of higher rates of paravalvular regurgitation [3]. Similar results have been published for the CV [11]. The improved hemodynamic performances observed with TAVI were unexpected because this procedure requires the crushing of the native severely calcified valve by the stented prosthesis—a mechanism which may be characterized by obstructive elements per se.

Our current findings demonstrate that all three types of valves showed excellent EOAs and Doppler hemodynamic performances both at short-term and at 1-year follow-up. While EOAs and prosthesis size were found to increase in parallel, the transvalvular gradients were comparable across different sizes of the same prosthesis. Size-specific EOAs observed in our study were similar to those previously reported in the literature for the S3 prosthesis [12], but larger than those of the CV [13]. However, Spethmann et al. [13] reported smaller EOAs for CV 26 mm and CV 29 mm (1.9 ± 0.53 cm^2^ and 2.1 ± 0.42 cm^2^, respectively). Such discrepancies may at least in part be explained by differences in the experimental design. Specifically, patients in the study by Spethmann et al. [13] were followed up prospectively and LVOT diameters were accurately measured in the pre-TAVI phase by transesophageal echocardiography.

In this study, we found larger EOAs for self-expandable valves compared with balloon-expandable valves implanted in patients with small aortic annulus (i.e, S3 23 mm versus CV 26 mm and EV 26 mm). Similar results have been reported in a case-matched study in which MG values were higher and EOAs smaller for S3 compared with EVR, at the expense of an increased paravalvular regurgitation in patients implanted with the latter prosthesis [14].

At 1-year follow-up, we found similar rates of more than mild aortic regurgitation for S3 and EVR valves (5% and 4%, respectively). This finding indicates that the self-expandable CV may be clinically useful to prevent PPM in patients at high risk of mismatch occurrence [15].

Our results with regard to transvalvular gradients are in line with those obtained for S3 prostheses in the Partner [16] and SOURCE 3 registries [17] (11.6 mmHg versus 11.3 mmHg and 12.3 mmHg, respectively). Herrmann et al. [16] reported similar size-specific transvalvular gradients for S3 prostheses (i.e., 13 mmHg, 10.6 mmHg, and 9.4 mmHg for the 23 mm, 26 mm, and 29 mm valves, respectively). In the CV group, the mean gradients observed in our study were lower than those reported in the CoreValve US Trial [18] but comparable to those observed by Spethmann et al. [13] (7.3 mmHg versus 9 mmHg and 7.6 mmHg, respectively).

Few data are currently available on the hemodynamic performances of new-generation self-expandable EVR valves. Popma et al. [19] published highly similar values at 1-month follow-up in terms of both mean EOA (1.9 cm^2^ versus 2.1 cm^2^ in our cohort) and MG (7.8 mmHg versus 8.8 mmHg in our cohort).

Hahn et al. [20] have recently reported a series of hemodynamic parameters for different generations of balloon-expandable and self-expanding valves (according to both size and type). With regard to the three different sizes of S3 valves, we found similar values with respect to EOA and MG. As far as self-expanding prostheses are concerned, we found similar MG values but higher EOAs than those reported by Hahn and coworkers [20]. This apparent discrepancy may stem from the different method used to measure the EOAs of self-expanding valves. While Hahn et al. [20] used stroke volume (2D disc method) when the outer-to-outer border of the prosthesis was difficult to measure, EOA calculations in our study were based on the same pre-TAVI LVOT diameter.

Because valve manufacturers do not provide data on in vivo EOA, we believe that our study yields valuable information on valve hemodynamics following TAVI—which may in turn be clinically useful to assess valve function over time.

PPM has been previously associated with a reduced survival in patients implanted with surgical aortic valve bioprostheses [21] but not in those undergoing TAVI [22]—in whom this phenomenon is known to occur less frequently. Although the overall PPM rate observed in our cohort was lower than those described in most previous TAVI studies (mean published PPM prevalence: 31%) [22], our results are in keeping with the 11% rate recently reported by Külling et al. [12]. We believe that the reduction in the rate of PPM may result from both improved imaging modalities and a better experience in prosthesis sizing. 

With regard to all-cause mortality, we did not find significant differences between S3, CV, and EVR valves at a 1-year follow-up. The overall mortality rate observed in our cohort was similar to those previously reported for the three types of prostheses (ranging between 8.9 and 14.4%) [16,17,18,23].

Taken together, our findings indicate that the observed differences in EOAs and MG values between the three types of prosthesis did not have a significant impact on all-cause mortality in the first year following implantation. Larger studies with longer follow-up periods are required to investigate the long-term clinical impact of PPM in patients undergoing TAVI.

Our findings need to be interpreted in the context of some limitations. First, the retrospective collection of echocardiography data does not allow establishing whether the pulsed Doppler sample was invariably placed before the prosthesis stent. Consequently, EOAs could have been overestimated as a result of the flow acceleration phenomenon. Owing to the high variability of LVOT measurements, all EOAs were recalculated on consecutive examinations using the same pre-TAVI LVOT diameter. Second, echocardiography data at 6 and 12 months of follow-up were unavailable for 16.5% of the participants. Third, the study sample mainly consisted of patients who were implanted with the S3 23 mm and 26 mm prostheses (54% of study population), whereas other devices (specifically CV 23 mm and CV 31 mm) were used only in a small number of patients (ultimately hampering a reliable calculation of EOAs and transvalvular MG values in these subgroups). Fourth, no gold standard exists against which the calculation of the different size-specific EOAs can be compared. Finally, an operator-dependent selection bias for the selection of BEV versus SEV may exist. This may be at least in part dependent on clinical characteristics not collected in our dataset—including difficult ilio-femoral access (e.g., presence of calcifications or atheroma), aortic tortuosity, and extensive aortic annulus calcifications.

## 5. Conclusions

At short-term and at 1-year follow-up, both balloon-expandable and self-expandable prostheses are characterized by excellent hemodynamic performances (as assessed by EOAs and MG values)—without evidence of very early valve degeneration. In patients with small aortic annulus, self-expandable CV and EVR prostheses are characterized by significantly higher EOAs and lower MG values compared with balloon-expandable S3 valves—both at short-term and at 1-year follow-up. While these hemodynamic differences did not appear to have a major impact on overall mortality in our study, independent replication is necessary. Larger studies with longer follow-up periods are thus required to confirm and expand our findings.

## Figures and Tables

**Figure 1 jcm-10-00186-f001:**
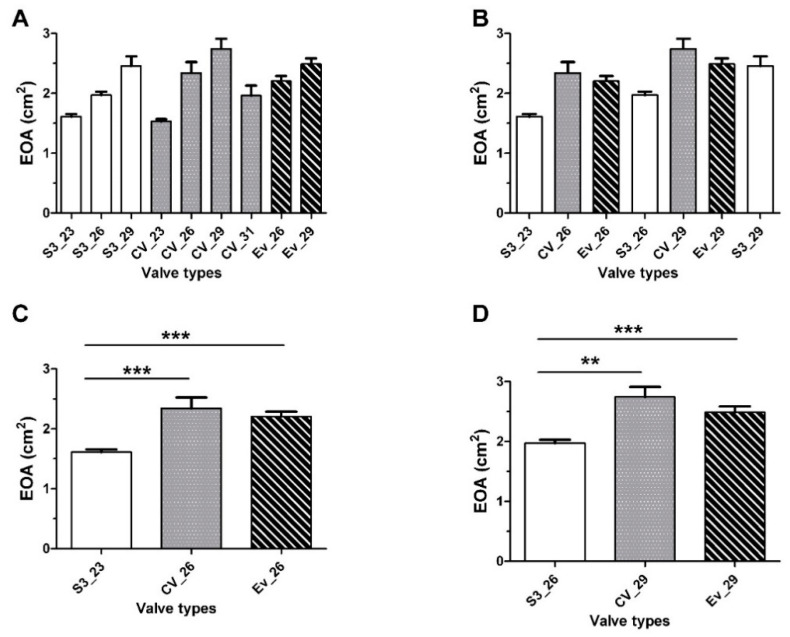
Effective orifice areas (EOAs; panels (**A**,**B**)) in patients implanted with Sapien S3^®^ 23 mm, 26 mm, or 29 mm valves (S3_23, S3_26, and S3_29, respectively), CoreValve^®^ 23 mm, 26 mm, 29 mm, or 31 mm valves (CV_23, CV_26, CV_29, and CV_31, respectively), or Evolut^®^ R 26 mm or 29 mm valves (EV_26 and EV_29, respectively). Panels (**C**,**D**) show the EOAs of the Sapien S3^®^, CoreValve^®^, and Evolut^®^ R prostheses in patients with similar aortic annulus sizes. ** *p* < 0.01 and *** *p* < 0.0001 for Sapien S3^®^ versus CoreValve^®^ or Sapien S3^®^ versus Evolut^®^ R.

**Figure 2 jcm-10-00186-f002:**
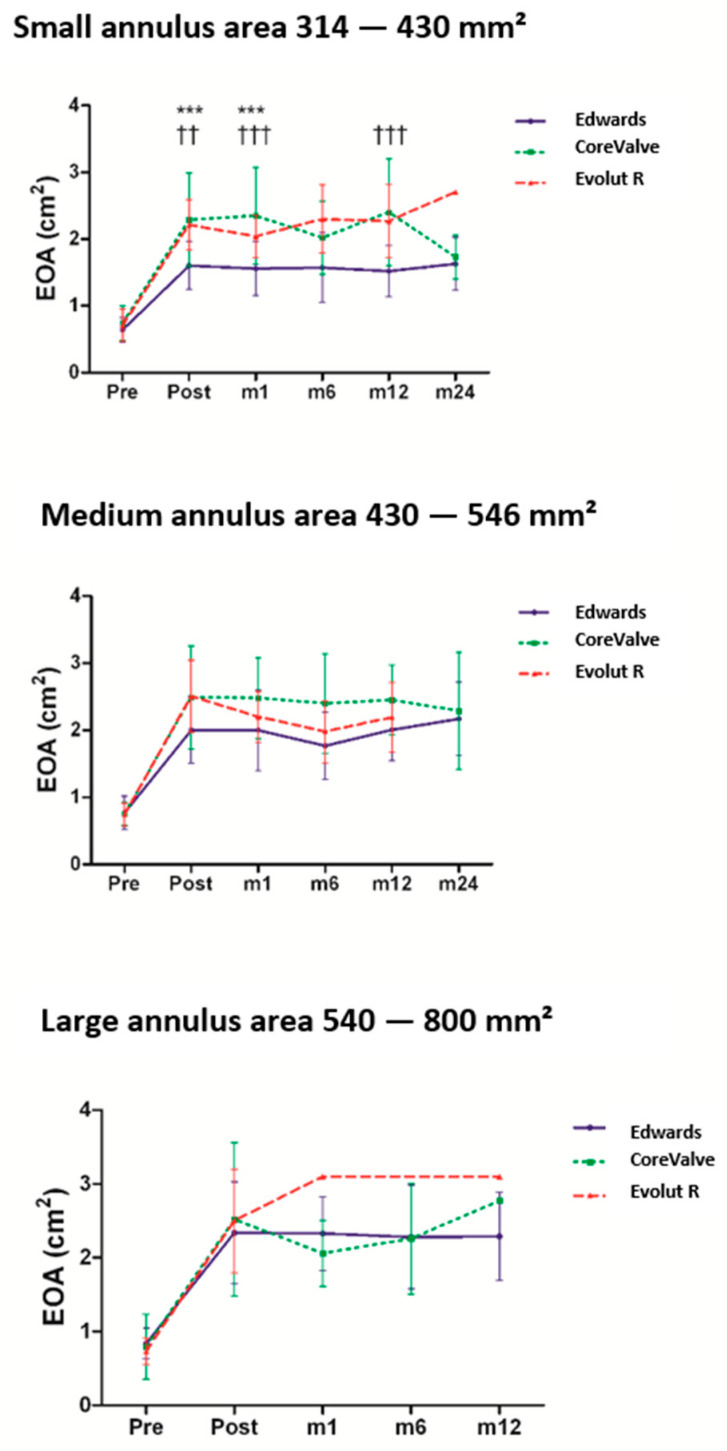
Temporal changes of effective orifice areas (EOAs) from the pre-implantation (pre) phase to the post-implantation (post) period—i.e., at 1, 6, 12, and 24 months (M1, M6, M12, and M24, respectively) of follow-up—in three groups of patients according to their aortic annulus area measured on CT images. *** *p* < 0.001 for Edwards Sapien S3^®^ versus CoreValve^®^. ††† *p* < 0.001 and †† *p* < 0.01 for Edwards Sapien S3^®^ versus Evolut^®^ R.

**Figure 3 jcm-10-00186-f003:**
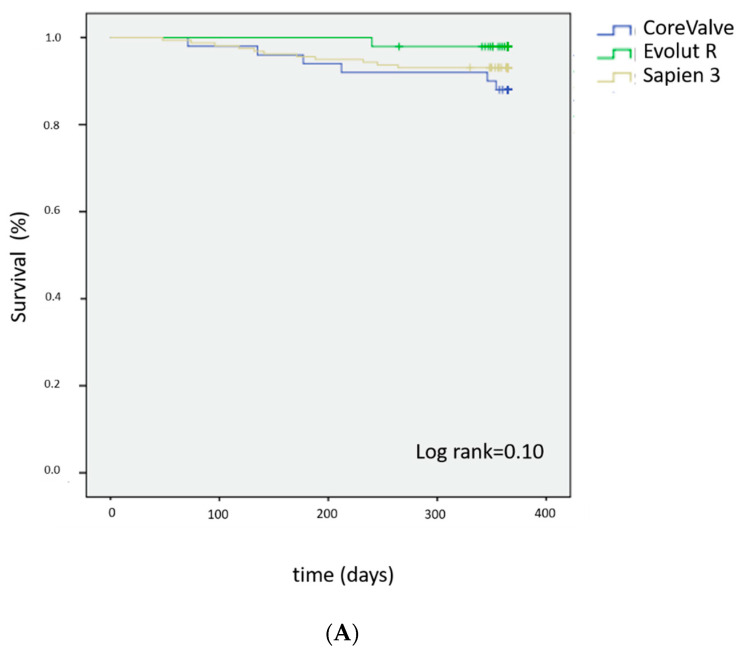
Kaplan–Meier plots of overall mortality in the intention-to-treat (panel (**A**)) and per-protocol (panel (**B**)) populations observed from hospital discharge to 1-year follow-up in patients implanted with the Sapien S3^®^, CoreValve^®^, and Evolut^®^ R prostheses.

**Table 1 jcm-10-00186-t001:** Baseline characteristics of the study patients (*n* = 260).

Valve Type	Overall	Sapien 3 (*n* = 161)			CoreValve (*n* = 50)				Evolut R (*n* = 49)		*p*
Diameter (mm)	(*n* = 260)	23 (*n* = 74)	26 (*n* = 67)	29 (*n* = 20)	23 (*n* = 2)	26(*n* = 15)	29 (*n* = 24)	31 (*n* = 9)	26 (*n* = 22)	29 (*n* = 27)	
Age (years)	84.1 ± 6.8	85.6 ± 6.5	84 ± 5.6	82.5 ± 6.2	88.9 ± 6.5	84.1 ± 8.4	83.3 ± 8.9	80.5 ± 7.6	84.4 ± 6.4	83.3 ± 7.5	0.347
Female sex, *n* (%)	146 (56.1)	65 (87.8)	21 (31.3)	1 (5)	1 (50)	13 (86.7)	11 (45.8)	1 (11.1)	20 (90.9)	13 (48.1)	<0.001
Euroscore I, %	22 ± 14	23.2 ± 13.2	21.8 ± 14.2	18.9 ± 10.4	17.5 ± 0.7	20.5 ± 10.9	23.8 ± 12.1	34.9 ± 32.7	17.7 ± 10.6	19.8± 12.5	0.12
Euroscore II, %	4.9 ± 5.9	4.7 ± 3.5	5.2 ± 8	3.9 ± 2.9	2.5 ± 0.6	3.3 ± 1.6	4.5 ± 2.2	14.2 ± 15.8	4 ± 2.7	4.2 ± 2.7	0.001
Body surface (m^2^)	1.8 ± 0.2	1.7 ± 0.19	1.85 ± 0.2	1.88 ± 0.19	1.59 ± 0.2	1.65 ± 0.1	1.92 ± 0.21	1.90 ± 0.1	1.77 ± 0.1	1.84 ± 0.1	<0.001
BMI (kg/m^2^)	26.3 ± 5.3	25.7 ± 6.2	26.9 ± 4.8	26.4 ± 3.7	22.3 ± 0.1	23.9 ± 5.9	28.5 ± 5.5	24.4 ± 3.4	26.6 ± 4.8	26.5 ± 3.7	0.186
Heart failure, *n* (%)	184 (70.8)	53 (71.6)	44 (65.7)	14 (70)	1 (50)	10 (66.7)	15 (62.5)	6 (66.7)	17 (77.3)	24 (88.9)	0.53
Diabetes, *n* (%)	92 (35.4)	25 (33.8)	24 (35.8)	2 (10)	0 (0)	3 (20)	14 (58.3)	3 (33.3)	11 (50)	10 (37)	0.042
Hypertension, *n* (%)	224 (86.2)	64 (86.5)	57 (85.1)	13 (65)	2 (100)	11 (73.3)	22 (91.7)	8 (88.9)	22 (100)	25 (92.6)	0.057
AFIB, *n* (%)	71 (27.3)	16 (21.6)	19 (28.4)	4 (20)	0 (0)	4 (26.7)	4 (16.7)	3 (33.3)	8 (36.4)	13 (48.1)	0.2
Pacemaker, *n* (%)	36 (13.8)	9 (12.2)	14 (20.9)	2 (10)	0 (0)	0 (0)	5 (20.8)	2 (22.2)	2 (9.1)	2 (7.4)	0.35
CKD, *n* (%)	140 (53.8)	35 (47.3)	31 (46.3)	11 (55)	2 (100)	7 (46.7)	16 (66.7)	7 (77.8)	14 (63.6)	17 (63)	0.24
PCI, *n* (%)	97 (37.3)	29 (39.2)	25 (37.3)	7 (35)	2 (100)	4 (26.7)	9 (37.5)	5 (55.6)	6 (27.3)	10 (37)	0.59
CABG, *n* (%)	25 (9.6)	10 (13.5)	4 (6)	1 (5)	0 (0)	0 (0)	2 (8.3)	2 (22.2)	1 (4.5)	5 (18.5)	0.3
Peripheral artery disease, *n* (%)	68 (26.2)	16 (21.6)	12 (17.9)	5 (25)	0 (0)	3 (20)	12 (50)	5 (55.6)	5 (22.7)	10 (37)	0.03
Stroke, *n* (%)	32 (12.3)	9 (12.2)	9 (13.4)	2 (10)	0 (0)	0 (0)	5 (16.7)	1 (11.1)	3 (13.6)	4 (14.8)	0.92
COPD, *n* (%)	23 (8.8)	6 (8.1)	8 (11.9)	2 (10)	0 (0)	0 (0)	1 (4.2)	1 (11.1)	2 (9.1)	3 (11.1)	0.9
Previous BAV, *n* (%)	27 (10.4)	9 (12.2)	7 (10.4)	5 (25)	0 (0)	0 (0)	3 (12.5)	0 (0)	2 (9.1)	1 (3.7)	0.31

Abbreviations: BMI, body mass index; AFIB, atrial fibrillation; CKD, chronic kidney disease; PCI, percutaneous coronary intervention; CABG, coronary artery by-pass graft; COPD, chronic obstructive pulmonary disease; BAV, balloon aortic valvuloplasty.

**Table 2 jcm-10-00186-t002:** Baseline characteristics of the study patients (*n* = 260).

Valve Type	Overall	Sapien 3 (*n* = 161)			CoreValve (*n* = 50)				Evolut R (*n* = 49)		*p*
Diameter (mm)	(*n* = 260)	23 (*n* = 74)	26 (*n* = 67)	29 (*n* = 20)	23 (*n* = 2)	26(*n* = 15)	29 (*n* = 24)	31 (*n* = 9)	26 (*n* = 22)	29 (*n* = 27)	
Echocardiography											
LVEF, %	53.8 ± 13.3	54.9 ± 12.9	49.9 ± 13.1	47.7 ± 11.8	68.5 ± 13.3	58.3 ± 14.6	54 ± 12.3	45.2 ± 19.4	60.4 ± 12.3	58.9 ± 8.7	<0.001
MG, mmHg	49.5 ± 15.1	52.7 ± 17	44.1 ± 13.1	48.8 ± 19.5	53 ± 24	55 ± 11.9	47.8 ± 12.4	41.5 ± 13.2	53.8 ± 13.3	51.9 ± 12.7	0.013
SVi, mL/m^2^	41.6 ± 11.3	40.1 ± 9.6	39.9 ± 11.2	44.9 ± 10.1	40 ± 4.2	44.3 ± 12.6	39.3 ± 9.6	39.3 ± 14	44.9 ± 13.4	46.4 ± 13.1	0.093
AVA, cm^2^	0.73 ± 0.22	0.62 ± 0.17	0.79 ± 0.22	0.87 ± 0.23	0.9 ± 0.67	0.66 ± 0.19	0.78 ± 0.14	0.81 ± 0.32	0.71 ± 0.24	0.76 ± 0.18	<0.001
AR > 2, *n* (%)	27 (10.4)	6 (8.1)	5 (7.4)	5 (25)	0 (0)	0 (0)	2 (8.3)	2 (22.2)	2 (9.1)	2 (7.4)	0.009
MR > 2, *n* (%)	33 (12.7)	14 (18.9)	5 (7.4)	5 (25)	0 (0)	4 (26.7)	2 (8.3)	1 (11.1)	0 (0)	2 (7.4)	0.049
LVOT, mm	21.3 ± 1.9	20.2 ± 1.4	21.9 ± 1.6	24 ± 1.8	21 ± 2.8	20 ± 1.7	22 ± 1.3	22.8 ± 1.6	20 ± 1.5	21.9 ± 1.4	<0.001
CT											
Area-derived diam	24 ± 2.1	22.4 ± 1.5	24.8 ± 1.5	27.4 ± 0.96	23	21.7 ± 1.3	25.2 ± 1.1	26.8 ± 1.2	22 ± 1.3	24 ± 1.2	<0.001
Ann area, mm^2^	453.3 ± 86.5	388.6 ± 44.7	488 ± 60.5	608.4 ± 53.5	420	391 ± 64.4	494 ± 45.5	545.1 ± 68	375.6 ± 46.2	460.2 ± 66.6	<0.001
Ann perim, mm	75.7 ± 6.7	70.2 ± 3.1	77.9 ± 8.5	90.7 ± 6.08	-	-	79.2 ± 1.7	83.7 ± 4.9	70.1 ± 4.5	77.1 ± 2.9	<0.001
Calcium score, AU	3383 ± 1682	2614 ± 1129	3392 ± 1529	5790 ± 2596	-	2957 ± 1642	3016 ± 989	4195 ± 1932	3384 ± 1503	3365 ± 1637	0.01

**Table 3 jcm-10-00186-t003:** Evolution of EOA, EOAi, MG, and LVEF values at follow-up.

Valve Type	Overall	Sapien S3			CoreValve				Evolut R	
Diameter (mm)		23	26	29	23	26	29	31	26	29
	*n* = 260	*n* = 74	*n* = 67	*n* = 20	*n* = 2	*n* = 15	*n* = 24	*n* = 9	*n* = 22	*n* = 27
EOA, cm^2^ (SD)										
Discharge	2.0 ± 0.6	1.6 ± 0.3	1.9 ± 0.4	2.4 ± 0.7	1.5	2.3 ± 0.7	2.7 ± 0.8	1.9 ± 0.5	2.2 ± 0.3	2.4 ± 0.4
1 month	2.0 ± 0.6	1.5 ± 0.4	2 ± 0.5	2.3 ± 0.5	1.7	2.3 ± 0.6	2.6 ± 0.5	1.9 ± 0.4	2 ± 0.3	2.2 ± 0.4
6 months	1.8 ± 0.6	1.4 ± 0.3	1.9 ± 0.6	2.3 ± 0.7	/	1.8 ± 0.6	2.5 ± 0.6	2 ± 0.7	2.1 ± 0.5	2.2 ± 0.5
1 year	2.0 ± 0.6	1.4 ± 0.3	2.1 ± 0.4	2.3 ± 0.6	/	2.1 ± 0.5	2.5 ± 0.5	2.4 ± 0.2	2.1 ± 0.4	2.5 ± 0.6
EOAi, cm^2^ (SD)										
Discharge	1.1 ± 0.3	0.9 ± 0.2	1 ± 0.2	1.3 ± 0.4	0.94	1.4 ± 0.4	1.51 ± 0.5	1.0 ± 0.3	1.2 ± 0.2	1.4 ± 0.3
1 month	1.1 ± 0.3	0.9 ± 0.2	1.1 ± 0.3	1.2 ± 0.3	1.25	1.3 ± 0.3	1.4 ± 0.3	1 ± 0.2	1.2 ± 0.2	1.2 ± 0.2
6 months	1 ± 0.3	0.8 ± 0.2	1 ± 0.3	1.2 ± 0.3	/	1 ± 0.3	1.3 ± 0.2	1.1 ± 0.4	1.1 ± 0.4	1.2 ± 0.2
1 year	1.1 ± 0.3	0.8 ± 0.2	1.1 ± 0.2	1.1 ± 0.3	/	1.2 ± 0.3	1.3 ± 0.2	1.2 ± 0.0	1.2 ± 0.2	1.3 ± 0.4
MG, mmHg (SD)										
Discharge	9.4 ± 3.9	11.7 ± 3.4 ^$^	9.5 ± 3.6 *	8.9 ± 2.9	14.5 ± 6.3	6.7 ± 3.9 ***	6.4 ± 3.2 *** ^$^	8.5 ± 4.7	8.7 ± 3.2 *	7.4 ± 3.5 ***
1 month	10 ± 4.3	12.4 ± 4.3	10.3 ± 3.7	10.5 ± 5.7	8	6.5 ± 2.6 *** ^$^	6.1 ± 2.3 *** ^$$$^	10.2 ± 5.1	8.4 ± 2.6 *	8.5 ± 2.8 ***
6 months	10.9 ± 4.3	13.5 ± 3.5 ^%^	11.1 ± 4.3	9.1 ± 3.6	/	7.8 ± 3.1 *	7 ± 3 * ^$^	10 ± 2.9	8.8 ± 4.2 *	9.2 ± 3.5
1 year	10.2 ± 4.7	14 ± 4.5	9.6 ± 3.6 ***	9.7 ± 3.7 *	/	6.9 ± 3.6 *	6.8 ± 2.5 ***	10 ± 3	8.3 ± 2.5 **	7.3 ± 4.3 ***
LVEF, %(SD)										
Discharge	56 ± 13	59 ± 12	52 ± 14	47 ± 14	69 ± 4	58 ± 15	58 ± 11	45 ± 15	65 ± 6	58 ± 9
1 month	57 ± 11	60 ± 11	54 ± 10	46 ± 12	66	61 ± 6	58 ± 8	45 ± 14	62 ± 7	59 ± 8
6 months	58 ± 11	60 ± 11	56 ± 9.5	49 ± 16	/	64 ± 6	59 ± 13°	49 ± 12	65 ± 7	59 ± 13
1 year	58 ± 11	60 ± 11	55 ± 10	51 ± 12	/	61 ± 11	62 ± 10	51 ± 16	64 ± 7	59 ± 12

Abbreviations: EOA, effective aortic area; EOAi, effective orifice area indexed to the body surface area; MG, transvalvular aortic mean gradient; LVEF, left ventricular ejection fraction. *****
*p* < 0.05, ******
*p* < 0.001, *******
*p* < 0.0001 versus Sapien S3 23 mm (Bonferroni’s post hoc test), ^$^
*p* < 0.05, ^$$$^
*p* < 0.0001 versus Sapien S3 26 mm (Bonferroni’s post hoc test), ^%^
*p* < 0.05, difference in mean gradient versus Sapien S3 at 1 month and 6 months (paired Student’s *t*-test).

## Data Availability

The data presented in this study are available on request from the corresponding author. The data are not publicly available due to anonymization issues.

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
