# Peer review of "Effective Orifice Area of Balloon-Expandable and Self-Expandable Transcatheter Aortic Valve Prostheses: An Echo Doppler Comparative Study"

_jcm, 2021, doi:10.3390/jcm10020186_

Round 1

Reviewer 1 Report

Thank you for addressing my questions and comments.

Author Response

We thank the reviewer for the positive feedback. 

Reviewer 2 Report

Thanks for the opportunity to review the manuscript.

Author Response

Thank you for your supportive comment. 

Reviewer 3 Report

Introduction

First paragraph is very generic and does not teach the reader anything new.

“However, there are few published data on the intra-individual hemodynamic changes that may occur over time in patients treated with new generation valves” – this data is readily available in both the Sapien and corevalve Intermediate and low risk trials

“We therefore designed the current study to investigate this issue by focusing on the EAO EOA and mean 51 transvalvular aortic gradient (MG) in relation to valve size.” – This does not follow the problem they have identified (see paragraph above)

Method

Single observer performing all studies, this is a strength of the study

Results:

Impact of prosthesis size on the hemodynamic characteristics

For me this is a meaningless comparison. The recommendations for which size valve to use differs from manufacturer to manufacturer, and selection of valves is also based on anatomy for the patient. To compare EOA/MG for two valves from different manufacturers head-to-head does not provide any meaningful information. I should remove this paragraph

The comparison made in the second paragraph is the only valid one, but interpretation of the findings should be done carefully.

The authors state that they did a Bonferroni post-hoc test, but present a lot of results with p<0.05

Paravalvular aortic regurgitation

In this type of study, it is reasonable to compare the different devices in PVL. RCTs on both the Corevalve/Evoloute and Sapien platform show that the Sapien platform has lower PVL that the Corevalve/Evolute platform.  As this study focuses on EOA/MG only, it does not give a fair comparison between the two platforms.

Survival analysis

Does not add anything to the study. Can be omitted.

Discussion

The authors reach quite far-reaching conclusion regarding the which valves perform the best. This is a retrospective study and they have presented few or none confounding factors. Did the Sapien group differ from the Corevalve/Evolute group? If they gave both platforms, how do they choose which valve to which patient? There must be a selection bias. Which valve do they use for bicuspids, heavily calcified leaflets, protruding calcium in the LVOT, younger patients, older patients, patients  with poor renal function? There must be a selection bias. In our department we have an algorithm for picking the right valve for the patient, so by definition we have selection bias. I would be very surprised if they do not have a selection bias in this cohort as well.

Author Response

AUTHORS’ RESPONSES TO COMMENTS FROM REVIEWER #3

Introduction

First paragraph is very generic and does not teach the reader anything new.

 “However, there are few published data on the intra-individual hemodynamic changes that may occur over time in patients treated with new generation valves” – this data is readily available in both the Sapien and corevalve Intermediate and low risk trials

Response: Thank you for your constructive criticisms. Using an institutional registry, Kulling et al. were the first to investigate the EOA of TAVI prostheses by focusing on the Sapien 3 devices (Külling et al. Effective orifice area and hemodynamic performance of the transcatheter Edwards Sapien 3 prosthesis: short-term and 1-year follow-up. Eur Heart J - Cardiovasc Imaging. 01-01 2018;19(1):23–30). On designing the current study, we sought to expand the available knowledge in a larger cohort of patients who were implanted with either balloon expandable (BEV) or self-expandable valves (SEV). All of the study patients were recruited from a large university hospital where TAVI procedures were routinely performed. While our study was being conducted, Hahn et al. published normative values – including EOA and hemodynamic parameters (e.g., mean gradient and Doppler velocity index) – for commercially available THVs according to valve type and size; these results were obtained in a selected population of patients who had participated in TAVI randomized controlled trials of BEV and SEV (Hahn et al. Comprehensive Echocardiographic Assessment of Normal Transcatheter Valve Function. JACC Cardiovasc Imaging. 01 2019;12(1):25–34). While we acknowledge that the topic of study is not entirely original and we cannot claim novelty, we believe that our data confirm and expand our knowledge on published reference values. Specifically, we show that they are reproducible in a real-world echo-laboratory practice. In line with the work by Hahn et al., we show data on EOA at discharge and at 30 days. However, we were also able to provide follow-up data at 1 year. These results may be valuable to assess THV deterioration over time.

“We therefore designed the current study to investigate this issue by focusing on the EOA and mean transvalvular aortic gradient (MG) in relation to valve size.” – This does not follow the problem they have identified (see paragraph above)

Response: Thank you for highlighting this. The purpose of our study was two-fold, i.e., 1) to investigate reference values of EOA and MG according to valve type and size and 2) to assess the evolution of these parameters over time (1-year follow-up). Thus, the text was revised as follows:

However, there are few published data about reference hemodynamic parameters and their individual changes that may occur over time in patients treated with new generation valves, i.e., the self-expandable Corevalve® (Medtronic, Minneapolis, MN, USA), the Corevalve Evolut-R® system (Medtronic), and the balloon-expandable Edwards Sapien 3® system (Edwards Lifesciences Inc., Irvine, CA, USA). We therefore designed the current study to investigate this issue by focusing on the EOA and mean transvalvular aortic gradient (MG) in relation to valve size. We also examined the evolution of these parameters at 30-day and 1-year follow-up.

Methods

Single observer performing all studies, this is a strength of the study

Response: We appreciate the supportive comment.

Results

Impact of prosthesis size on the hemodynamic characteristics

For me this is a meaningless comparison. The recommendations for which size valve to use differs from manufacturer to manufacturer, and selection of valves is also based on anatomy for the patient. To compare EOA/MG for two valves from different manufacturers head-to-head does not provide any meaningful information. I should remove this paragraph. The comparison made in the second paragraph is the only valid one, but interpretation of the findings should be done carefully.

Response: Thank you for the pertinent comment. As suggested, we removed the paragraph mentioned by the reviewer from the “Results” section. In addition, Figure 2 was deleted.

The authors state that they did a Bonferroni post-hoc test, but present a lot of results with p<0.05

Response: Thank you for the valued comment. EAO and MG values were compared across three patient groups using ANOVA. Post-hoc Bonferroni tests were applied solely when the global ANOVA test produced a p value<0.05. Bonferroni tests allow identifying significant differences in pairwise group comparisons when the p value from ANOVA is statistically significant. Since the test was repeated three folds at each time point, the threshold for statistical significance was adjusted by applying the familywise error rate (FWER) correction, as follows: p<0.05/3=0.016. We corrected the corresponding figure (Figure 2 in the revised manuscript); consequently, only p value<0.001 and 0.0001 were considered statistically significant.

Paravalvular aortic regurgitation

In this type of study, it is reasonable to compare the different devices in PVL. RCTs on both the Corevalve/Evoloute and Sapien platform show that the Sapien platform has lower PVL that the Corevalve/Evolute platform.  As this study focuses on EOA/MG only, it does not give a fair comparison between the two platforms.

Response: The rate of PVL is reported in the “Results” section, paragraph 3.3. We concur that the S3 platform is the reference TAVR device in terms of PVL risk reduction. In our dataset, the rate of moderate paravalvular aortic regurgitation at discharge was 11.9% –without significant intergroup differences. It is conceivable that the limited sample size in our study did not allow achieving a sufficient statistical power to identify significant intergroup differences in terms of PVL.

Survival analysis

Does not add anything to the study. Can be omitted.

Response: Thank you for your comment. While the main goal of our study was to analyze the hemodynamic features of TAVR prosthesis over time, we believe that survival analysis may provide a valuable proxy for the clinical impact of hemodynamic findings. Of note, previous papers in the filed – including those authored by Kulling et al. and Hahn et al. – did not relate hemodynamic findings to survival figures. Thus, we wish to maintain this information in our manuscript.

Discussion

The authors reach quite far-reaching conclusion regarding the which valves perform the best. This is a retrospective study and they have presented few or none confounding factors. Did the Sapien group differ from the Corevalve/Evolute group? If they gave both platforms, how do they choose which valve to which patient? There must be a selection bias. Which valve do they use for bicuspids, heavily calcified leaflets, protruding calcium in the LVOT, younger patients, older patients, patients  with poor renal function? There must be a selection bias. In our department we have an algorithm for picking the right valve for the patient, so by definition we have selection bias. I would be very surprised if they do not have a selection bias in this cohort as well.

Response: We thank the reviewer for the constructive criticisms. We concur that – in a real-world setting – cathlab operators do routinely select the TAVR platform according to the patient’s characteristics. Specifically, we tended to select self-expandable devices in patients with difficult femoral access (e.g., presence of calcifications, tortuosity, ilio-femoral atheroma) and, more specifically, when aortic annulus calcifications are present. We indeed believe that SEV carry a lower risk of annulus rupture. Our dataset does not include information on these variables; however, the presence of PAD was over-represented in the SEV groups (Table 1). The limitation inherent to a possible selection bias for the implantation of BEV or SEV was acknowledged in the “Discussion” section, as follow: “Finally, an operator-dependent selection bias for the selection of BEV versus SEV may exist. This may be at least in part dependent on clinical characteristics not collected in our dataset – including difficult ilio-femoral access (e.g., presence of calcifications or atheroma), aortic tortuosity, and extensive aortic annulus calcifications”.

This manuscript is a resubmission of an earlier submission. The following is a list of the peer review reports and author responses from that submission.

Round 1

Reviewer 1 Report

In this manuscript, the authors assess size-specific effective orifice area of 3 prosthetic valve models after TAVI in patients with severe aortic stenosis, using doppler echocardiography at discharge, short-term, and 1-year followup in 260 patients. While they found improved hemodynamic performance of two models, none had any impact on all-cause mortality, nor was there evidence of valve degeneration.

-By what criteria was it determined that the “measured LVOT during follow-up was different from pre-TAVI LVOT value”? The continuity equation is highly sensitive to measures of LVOT; use of pre-implantation LVOT would appear to discount any changes caused by the implanted THV itself, or due to cardiac remodeling post-TAVR.

-Was the LVOT diameter measured at the same point as the LVOT VTI (immediately before the prosthesis), or at the same point in all patients? Presumably, structural differences in the S3, CV, and EVR would make placement of the VTI measurement (immediately before the prosthesis) different between valve models, and the LVOT diameter measurement would need to move in a corresponding manner?

-Did rates of mild-to-moderate paravalvular regurgitation differ between prostheses?

-Please review the text carefully for minor typos and grammatical issues (e.g. typo in the second line of the abstract (EAO vs EOA)

-The authors note their efforts to increase sample sizes by screening an additional cohort of 86 patients;  were power calculations performed for the statistical comparisons made in this study? If so, please provide these data.

-Were all TAVIs performed on tricuspid aortic valves, or did any patients have BAVs?

Reviewer 2 Report

Comments:

Please report the number of patients undergoing TTE at each time

The authors should compare TTE outcomes in patients with the same aortic annulus range.

I would omit data on CoreValve

Reviewer 3 Report

Comments to the authors

Thank you for the opportunity to review the manuscript titled “Effective orifice area of balloon-expandable and self-expandable transcatheter aortic valve prostheses: an echo Doppler comparative study” for the Journal of Clinical Medicine. The authors investigate the EOA and mean transvalvular aortic gradient (MG) in patients with new generation valves, including SAPIEN 3, Corevalve, and EvolutR. I agree that high transvalvular gradient and PPM after TAVR remain concerns because younger patients at lower surgical risk will be increasingly treated by TAVR and PPM would negatively affect their clinical outcomes. However, there are multiple issues to be considered.

  1. The main results of this manuscript include the followings. However, there isn’t anything new in the results.
    • Smaller valves had smaller EOAs regardless of the type of prosthesis.
    • In patients with the same aortic annulus size, hemodynamic performances of the self-expanding THV were superior to the balloon-expandable THV.
    • No difference was observed in mortality at one-year between S3, CV, and EVR valve.
  1. Some studies have been already published regarding 2-2. (1-4)

  1. Regarding the mortality at one-year, the cumulative morality in the Kaplan-Meier analysis was not evaluated multivariately, but univariately. Therefore, the analysis used in this study could not show an association between the EOA/MG and morality. The authors mentioned, “such hemodynamic differences did not have a significant impact on overall mortality” in the conclusion section. The statistical method is inappropriate to lead the conclusion.

  1. I found some “EAO” instead of “EOA”. Please correct them.

References

  1. Lanz J, Kim WK, Walther T, Burgdorf C, Mollmann H, Linke A, et al. Safety and efficacy of a self-expanding versus a balloon-expandable bioprosthesis for transcatheter aortic valve replacement in patients with symptomatic severe aortic stenosis: a randomised non-inferiority trial. Lancet. 2019;394(10209):1619-28.
  2. Okuno T, Khan F, Asami M, Praz F, Heg D, Winkel MG, et al. Prosthesis-Patient Mismatch Following Transcatheter Aortic Valve Replacement With Supra-Annular and Intra-Annular Prostheses. JACC Cardiovasc Interv. 2019;12(21):2173-82.
  3. Dvir D, Webb JG, Bleiziffer S, Pasic M, Waksman R, Kodali S, et al. Transcatheter aortic valve implantation in failed bioprosthetic surgical valves. JAMA. 2014;312(2):162-70.
  4. Hahn RT, Leipsic J, Douglas PS, Jaber WA, Weissman NJ, Pibarot P, et al. Comprehensive Echocardiographic Assessment of Normal Transcatheter Valve Function. JACC Cardiovasc Imaging. 2019;12(1):25-34.